# Fertiliser Effect of Ammonia Recovered from Anaerobically Digested Orange Peel Using Gas-Permeable Membranes

Carmo Horta [1,2,*], Berta Riaño [3], Ofélia Anjos [1,4,5] and María Cruz García-González [3]

1 Polytechnic Institute of Castelo Branco, School of Agriculture, Quinta da Sra. de Mércules, 6001-909 Castelo Branco, Portugal; ofelia@ipcb.pt
2 CERNAS-IPCB Research Centre for Natural Resources, Environment and Society, Polytechnic Institute of Castelo Branco, 6001-909 Castelo Branco, Portugal
3 Agricultural Technological Institute of Castilla y Léon. Ctra. Burgos, km. 119, 47071 Valladolid, Spain; riairabe@itacyl.es (B.R.); gargonmi@itacyl.es (M.C.G.-G.)
4 Centro de Estudos Florestais, Instituto Superior de Agronomia, Universidade de Lisboa, Tapada da Ajuda, 1349-017 Lisboa, Portugal
5 Centro de Biotecnologia de Plantas da Beira Interior, Quinta da Sra. de Mércules, 6001-909 Castelo Branco, Portugal
* Correspondence: carmoh@ipcb.pt

**Abstract:** The manufacture of mineral N fertilisers by the Haber–Bosch process is highly energy-consuming. The nutrient recovery technologies from wastes through low-cost processes will improve the sustainability of the agricultural systems. This work aimed to assess the suitability of the gas-permeable membrane (GPM) technology to recover N from an anaerobic digestate and test the agronomic behaviour of the ammonium sulphate solution (ASS) obtained. About 62% of the total ammonia nitrogen removed from digestate using GPM was recovered, producing an ASS with $14,889 \pm 2324$ mg N $L^{-1}$, which was more than six-fold higher than in digestate. The ASS agronomic behaviour was evaluated by a pot experiment with triticale as a plant test for 34 days in a growth chamber. Compared with the triticale fertilised with the Hoagland solution (Hoag), the ASS provided significantly higher biomass production (+29% dry matter), N uptake (+22%), and higher N agronomic efficiency 3.80 compared with 1.81 mg DM $mg^{-1}$N in Hoag, and a nitrogen fertiliser replacement value of 133%. These increases can be due to a biostimulant effect provided by the organic compounds of the ASS as assessed by the FT-Raman spectroscopy. The ASS can be considered a bio-based mineral N fertiliser with a biostimulant effect.

**Keywords:** bio-based fertiliser; FT-Raman; nitrogen fertiliser replacement value; orange bio-waste; plant biostimulant





## 1. Introduction

Nitrogen (N) is one of the essential nutrients for all living organisms, which needs to be applied to soil to overcome its natural limitation regarding the N crop needs. Usually, N is added to the soil using mineral fertilisers obtained from capturing the atmospheric $N_2$ by the high energy-consuming Haber–Bosch process. To capture the atmospheric $N_2$ into $NH_3$, the Haber–Bosch process has an energy footprint of 12.1 kWh $kg^{-1}$ of $NH_3$-N, corresponding to around 1% of the world's energy consumption [1]. The challenges in meeting the increasing worldwide N demand of $1.2 \times 10^{11}$ kg of $NH_3$-N $year^{-1}$ puts pressure on developing and optimising technologies for N recovery [2]. In this sense, the recovery of N from wastewater could partially offset the demand for nitrogen-based fertilisers [2]. The effluents produced after anaerobic digestion of agro-industrial waste (digestate) are an important source of N. At the same time, decreasing the N content from anaerobic digestate can reduce the environmental risks associated with its use as fertiliser in nitrate-vulnerable zones [3]. The current technologies for N recovery include precipitation

by struvite, ultrafiltration/ion exchange, ultrafiltration/reverse osmosis, or acid absorption following separation by gas stripping or gas-permeable membrane (GPM) technology [2,4]. Among those, the GPM technology is considered one of the best techniques for recovering N from waste streams [2].

Galloway and Cowling [5] illustrated the losses of N from the Haber–Bosch process through the food chain, from farm to fork, suggesting that for every 100 atoms of N incorporated into the N fertiliser, only 14 were consumed in a vegetarian diet and only 4 in a carnivorous diet. The remaining N is mainly lost by volatilisation ($NO_x$, $NH_x$, $N_2$), soil leaching or drainage waters, or incorporated into effluents from anthropogenic activities, agro-food industries, or livestock production. Therefore, recovering the N incorporated into these effluents contributes to saving natural resources (fossil fuels) and helps recover a nutrient into a bio-based mineral fertiliser with a well-known N concentration. Thus, some disadvantages of fertilising with bio-wastes (e.g., nutrients imbalance regarding crop needs, and variable N content) are overcome, and in addition the N recovery process also contributes to the circular bio-based economy.

Few studies have evaluated the use of N-rich solutions from bio-wastes. Sigurnjak et al. [6,7] and Vaneeckhaute et al. [8] tested N-rich solutions obtained from an air scrubber, and Majd [9] tested an N-rich solution from the GPM technology from liquid dairy manure. The ammonium sulphate solution (ASS) used in this work was obtained from the GPM technology applied to an anaerobic digestate provided by the co-digestion of an orange peel effluent mixed with swine manure. To the best of our knowledge, this is the first work in which the agronomic behaviour of the ASS from anaerobic digestate was tested that also included the orange peel in its composition. Indeed, the agronomic role of digestates from the anaerobic digestion are well known, not only acting as soil amendments increasing the level of the soil organic matter (SOM) [10–14] but also their role as a source of nutrients for crops [15–19]. However, the GPM technology can constitute another approach to obtain a bio-based liquid fertiliser with a known N concentration and with commercial value. Regarding the composition of the ASS, Riaño et al. [20] and García-González et al. [21] observed that some organic matter can also be recovered by GPM technology. So, the presence of organic compounds in the ASS obtained in this work can also be expected. However, the composition of the orange peel bio-waste can add value to the digestate and to the ASS since it contains tannins, phenolic compounds, carotenoids, dietary fibres, sugars, and organic acids (e.g., oxalic, citric, acetic, and succinic), making this bio-waste valuable for the recovery of bioactive compounds [22–25]. This work hypothesised that the GPM technology in addition to recover N from the anaerobic digestate can also recover some organic compounds, which can contribute to the improvement of the fertilising value of the ASS.

Therefore, the objective of this work was (i) to evaluate the N recovery efficiency from anaerobic digestate applying the gas-permeable technology and (ii) to evaluate the fertiliser value of the ASS as a liquid N fertiliser, in a pot experiment using as plant test the triticale (×*Triticosecale* Wittmack, var. Misionero) (iii); the early chemical characterisation of the recovered ASS was performed using FT-Raman spectroscopy.

## 2. Materials and Methods

### 2.1. Experimental Procedure for N Extraction

A batch experiment was conducted in 2 L wastewater vessels consisting of polyethene terephthalate (PET) plastic jars for an effective digestate volume of 1.5 L (Figure 1). The acid tank consisted of 500 mL Erlenmeyer flasks containing 100 mL of 1 N $H_2SO_4$ (Panreac). A peristaltic pump (Pumpdrive 5001, Heidolph, Schwabach, Germany) continuously recirculates the acidic solution through tubular membranes inside the digestate vessels and back into the acid tank using a constant flow rate of 12 L $d^{-1}$. The pH in the acidic tank increased as the membrane captured $NH_3$; therefore, the acid pH was adjusted to keep it below 2. Gas-permeable tubular membrane (48 cm long, 5.2 mm outer diameter) made of expanded polytetrafluoroethylene (e-PTFE) (Zeus Industrial Products Inc., Orangeburg, SC, USA)

was used for $NH_3$ capture. The membrane was submerged in digestate contained in PET jars, which were kept closed but not airtight. Ports were installed on top of the vessels to obtain samples and monitor pH and temperature. According to previous work, low-rate aeration was used to naturally increase digestate pH without chemicals [21]. Air was supplied using an aquarium air pump (Hailea, Aco-2201) from the bottom of the reactor through a porous stone. The airflow rate was controlled at 0.24 $L_{-air}$ L digestate$^{-1}$ min$^{-1}$ in the first two days and at 0.13 $L_{-air}$ L digestate$^{-1}$ min$^{-1}$ to the end of the experiment using an airflow meter (Aalborg, Orangeburg, NY, USA). The digestate was continuously agitated using magnetic stirrers.

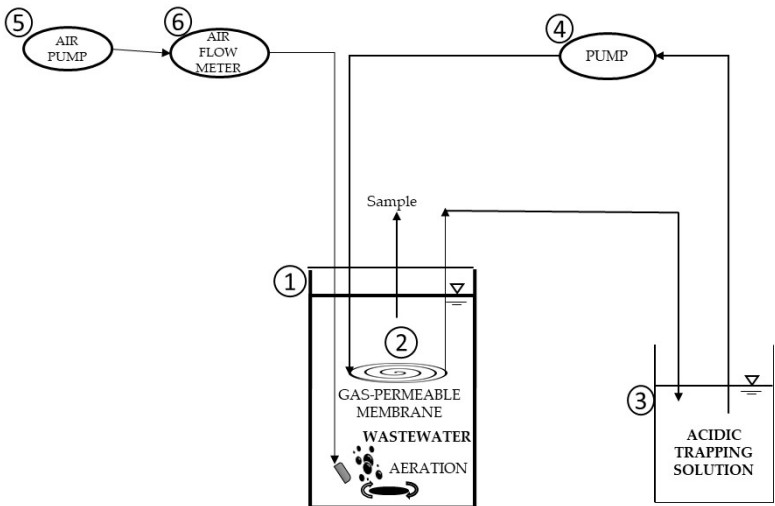

**Figure 1.** Experimental set-up of the gas-permeable membrane system, consisting of ① a wastewater vessel, ② a tubular membrane submerged in the wastewater, ③ a tank containing the acidic trapping solution, ④ a peristaltic pump that continuously recirculated the acidic solution through the tubular membrane, ⑤ an air pump to supply air to wastewater, and ⑥ an airflow meter to control the supplied airflow rate.

The experiment was carried out to evaluate N recovery from anaerobic digestion. This digestate was provided by the Centre for Scientific and Technological Research of Extremadura (CICYTEX) (Spain), and it was produced during the anaerobic co-digestion of orange peel with swine manure in mesophilic digesters operating at batch conditions for 73 days. Chemical characteristics are shown in Table 1.

**Table 1.** Chemical characterisation of the digestate at the beginning and at the end of the experiment.

|  | **Initial** | **Final** |
|---|---|---|
| Alkalinity (mg CaCO$_3$ L$^{-1}$) | 13,880 | 7431 |
| TS (g L$^{-1}$) | 35.4 ± 0.7 | 25.5 ± 3.5 |
| VS (g L$^{-1}$) | 22.3 ± 0.7 | 15.9 ± 2.3 |
| TCOD (mg L$^{-1}$) | 35,124 ± 847 | 22,866 ± 357 |
| SCOD (mg L$^{-1}$) | 3196 ± 171 | 4323 ± 342 |
| TKN (mg N L$^{-1}$) | 3474 ± 197 | 1823 ± 139 |
| TAN (mg N L$^{-1}$) | 2462 ± 47 | 1019 ± 24 |
| Pt (mg L$^{-1}$) | 1022 ± 1 | 541 ± 153 |

Digestate samples from the vessels and acidic solution samples from the acid tank were collected daily to monitor pH and total ammonia nitrogen (TAN). In addition, initial and final samples from the digestate were taken to determine total solid (TS), volatile solid (VS), total chemical oxygen demand (TCOD), soluble COD (SCOD), total Kjeldahl nitrogen (TKN), total phosphorous (Pt), and alkalinity. Evaluation of total chemical oxygen demand (TCOD) was also carried out in the recovered ammonium sulphate solution.

The experiment was duplicated twice, and results were expressed as means. The test was run at room temperature ($22.6 \pm 0.9$ °C) to simulate ammonia recovery from the digestate in a system external to the anaerobic digestion.

### 2.2. Experimental Procedure for Fertiliser Assessment

To evaluate the agronomic value of the ASS an experiment, was performed using triticale ($\times$ *Triticosecale* Wittmack, var. Misionero) as the crop test. It was performed in a micro-pot experiment since the amount of the ASS was too low, and it was impossible to perform a field or a pot experiment with a higher soil amount. The experiment was carried out in a growth climate chamber (aralab fitoclima 5000 PLH, Rio de Mouro, Portugal) at a temperature of 23 °C daytime and 12 °C night-time with 75% relative humidity and 16 h of daylight (90 mmol photons $m^{-2}s^{-1}$). Four treatments, each with four replicates, totalling 16 pots, were prepared. A mixture of 150 g of soil sieved at 2 mm with 50 g of purified sand was added to each pot. Three seeds of triticale were placed in each pot. The soil was kept at 75% of the water field capacity. After sprouting, two plants were left in each pot. A half-strength Hoagland solution with N (Hoag treatment) or without N (Control treatment—C) or the half-strength Hoagland solution without N plus the ASS with N recovered (ASS treatment) were applied in every pot/treatment during the experiment (totalling 100 mg of N for each pot in the Hoag and ASS treatments). The N sources for the triticale were the mineral N from the Hoagland solution or the N recovered from the digestate. Treatment without any fertilisation (W) was also carried out (Table 2).

**Table 2.** Treatments and fertilisation carried out in the micro-pot experiment.

| Treatments | Type of N Fertilisation | Applied N | [1] Other Nutrients |
|---|---|---|---|
| H$_2$O (W) | 0 | 0 | 0 |
| Control (C) | Hoagland without N | 0 | yes |
| Hoagland (Hoag) | Hoagland with N | [2] 100 mg per pot | yes |
| ASS | ASS + Hoag. without N | [2] 100 mg per pot | yes |

[1] nutrients applied by the half-strength Hoagland solution; [2] each pot had two triticale plants.

The soil used in the experiment was a dystric Regosol [26], sandy-loam (75% sand, 14% silt and 11% clay), slightly acidic (pH = 6.1), with low salinity (EC = 0.19 dS $m^{-1}$, 1:2 soil:water ratio), high in organic matter (OM = 4.7%) and with 2.96 g $kg^{-1}$ of total nitrogen, medium level of available P (AL-P = 34 mg $kg^{-1}$), high level of available K (AL-K = 380 mg $kg^{-1}$), low cation exchange capacity (6.5 cmol$_c$ $kg^{-1}$), and low levels of exchangeable Ca (4.6 cmol$_c$ $kg^{-1}$) and Mg (1.2 cmol$_c$ $kg^{-1}$). The half-strength Hoagland solution and the ASS were adjusted to pH 5.5 ($\pm$0.1) with 0.1 M KOH solution. The pots were watered daily with deionised water to 75% of the field capacity. In order to provide a continuous nitrogen supply to crop, the fertilisation was carried out continuously throughout the experiment. Therefore, the N fertilisation was spread over 18 applications to obtain a total amount of 100 mg N per pot at the end of the experiment. In each N fertilisation 13 mL $pot^{-1}$ of the Hoagland solution, a 3.3 mL $pot^{-1}$ of the ASS solution was applied. Then all the pots were watered with deionised water to 75% of the field capacity. The control of the amount of deionised water applied was carried out by the weight of the pots. The experiment duration was 34 days after sprouting, from 19 January to 21 February 2022. At the end of the experiment, the plants were cut, and the biomass and the total N were evaluated.

### 2.3. Analytical Methods

In the digestate, the concentrations of TS, VS, TCOD, SCOD, TKN, TAN, alkalinity, and Pt were analysed in accordance with the APHA methods [27]. The pH was measured using a pH-meter Crison Basic 20 (Crison Instruments S.A., Barcelona, Spain).

The ASS was analysed for total N by the Kjeldahl procedure following the APHA methods [27], and the pH was measured using a pH-meter Crison micropH 2002 (Crison Instruments S.A., Barcelona, Spain).

The triticale plants were cut and weighed to quantify biomass production (fresh matter). Then they were dried at 65 °C for 48 h and weighed again to quantify the yield on a dry matter (DM) production basis. Nitrogen was quantified by the Kjeldahl procedure (Nk), and the result was expressed on a DM basis. The reagents used in the analytical methods were Merck with a percentage of purity $\geq$ 98%. The nitrogen use efficiency (NUE) by the triticale was evaluated by different indices such as the N recovery efficiency (NRE), the N agronomic efficiency (NAE), the N replacement use efficiency (NRUE; %), and the N fertiliser replacement value (NFRV; %). These indices were calculated using Equation (1) to Equation (4):

$$\text{N recovery efficiency (NRE, \%)} = \left[\frac{(\text{N uptake} - \text{N0 uptake})}{\text{FN}}\right] \times 100 \tag{1}$$

$$\text{N agronomic efficiency}\left(\text{NAE; g DM g}^{-1}\text{N}\right) = \left[\frac{(\text{DMYN} - \text{DMY0})}{\text{FN}}\right] \tag{2}$$

$$\text{N replacement use efficiency (NRUE, \%)} = \frac{\frac{\text{N uptake ASS}}{\text{Total N applied ASS}}}{\frac{\text{N uptake Hoagland}}{\text{Total N applied Hoagland}}} \tag{3}$$

$$\text{N fertiliser replacement value (NFRV, \%)} = \frac{\frac{\text{N uptake ASS} - \text{N0 uptake}}{\text{Total N applied ASS}}}{\frac{\text{N uptake Hoagland} - \text{N0 uptake}}{\text{Total N applied Hoagland}}} \tag{4}$$

N uptake (g N pot$^{-1}$) is the amount of N uptake by the triticale in the N-fertilised pots, and N0 uptake is the amount of N uptake in the control pot (C treatment, without N fertilisation). The DMYN (g DM pot$^{-1}$) is the dry matter yield of the triticale in the N-fertilised pots, and DMY0 is the dry matter yield in the control pot. FN is the amount of N applied (g N pot$^{-1}$).

The relative yield was calculated by the following Equation (5), in which the biomass is the fresh matter or the dry matter production (g pot$^{-1}$):

$$\text{Relative yield (RY; \%)} = \left[\frac{\text{Biomass treatment}}{\text{Biomass Hoagland}}\right] \times 100 \tag{5}$$

*2.4. FT-Raman Analysis*

The spectral information of the liquid N fertilisers samples (ASS, Hoag and Control) was obtained using a FT-Raman spectrometer (BRUKER, MultiRAM) equipped with a 180° high-throughput collecting lens, an ultra-high sensitivity Liquid Nitrogen cooled Ge Diode detector, and an integrated 1064 nm, diode pumped, Nd:YAG laser with a maximum output power of 500 mW. The spectra were collected with 100 scans per spectrum at a spectral resolution of 4 cm$^{-1}$, with a scanner velocity of 5 kHz in the wavenumber range from 3500 to 200 cm$^{-1}$. Two measurements were performed for each sample in a quartz cell of 5 mm of optic space with the opposite face mirrored. The spectra were collected at controlled room temperature.

*2.5. Statistical Analysis*

One-way ANOVA analyses were conducted to identify the effect of the treatments (water, control, N-Hoagland solution, and ASS) on triticale biomass production and the N use efficiency indices. Tukey's test was used to compare means at a 0.05 probability level. All statistics were performed with IBM SPSS statistics 26.0 software (IBM SPSS

Statistics for Windows, Version 26.0, Armonk, NY, USA). For spectral data, analysis OPUS®, version: 7.5.18 (Bruker Optik, Ettlingen, Germany) was used.

## 3. Results and Discussion

### 3.1. N Recovery from the Digestate

The aeration increased the pH of the digestate from 8.48 to up 9.48 (Figure 2). TAN concentration in the digestate decreased from $2462 \pm 24$ to $1019 \pm 12$ mg N L$^{-1}$ within 4 days of the experiment (Figure 3). Simultaneously, the TAN concentration in the acidic solution increased to $14,889 \pm 2324$ mg N L$^{-1}$, which was more than six-fold higher than raw digestate. Sixty-two per cent of the TAN removed from digestate was recovered in the acidic solution. This value was the same as that obtained by García-González et al. [28] when recovering ammonia from anaerobically digestate manure using gas-permeable membranes in a 32-day experiment. A percentage of the TAN removed (37.7%) was volatilised to the atmosphere due to the high pH achieved during the treatment.

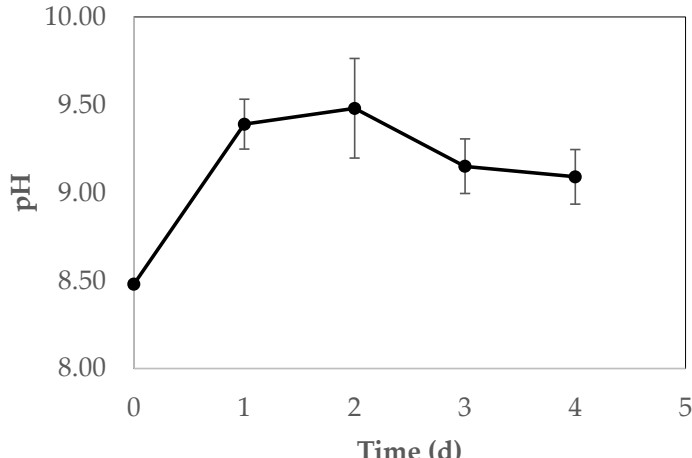

**Figure 2.** Evolution of pH in the digestate.

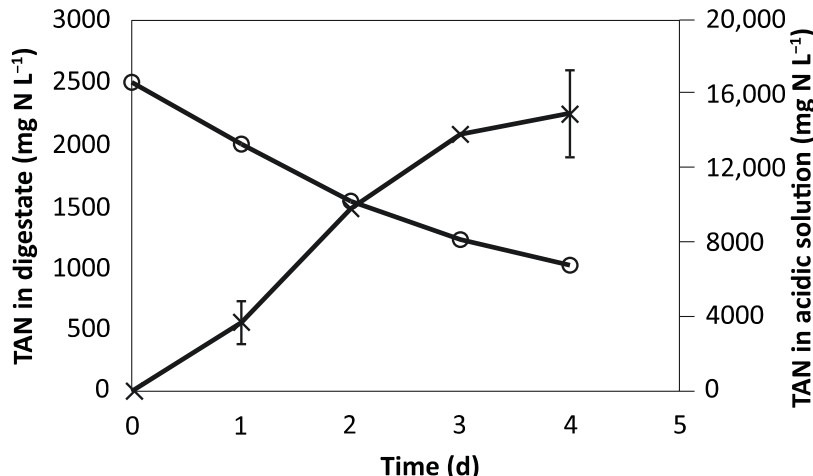

**Figure 3.** Evolution of TAN concentration in digestate (0) and in the acidic solution (x).

The TAN recovery was not linear and followed a 2nd-order curve, as shown in Figure 4, meaning that the TAN capture rate was higher during the first days and decreased with time. Specifically, this occurred after 3rd day of the experiment. The TAN recovery rate was $43.0 \pm 6.6$ g N per m$^2$ of membrane and per day. This value is higher than the values obtained in previous batch studies with similar treatment times found in the literature. For example, Vanotti et al. [29] treated anaerobically digested swine wastewater

containing 2350 mg N L$^{-1}$ using submerged membranes plus low-rate aeration to recover ammonia and reported a TAN recovery rate of 25.1 g TAN m$^{-2}$ d$^{-1}$. Dube et al. [30] tested gas-permeable membranes for recovery ammonia from anaerobically digested swine manure and obtained TAN recovery efficiencies between 22.7 and 30.7 g N m$^{-2}$ d$^{-1}$. The higher TAN recovery rate obtained in the present study could be attributed to the high pH (near 9.50) achieved in the digestate, and that it is the most critical parameter determining the amount of free NH$_3$ to pass through the membrane [31]. Other factors that can influence the TAN recovery rate by these membrane systems are the initial TAN concentration and the mixing [2,31].

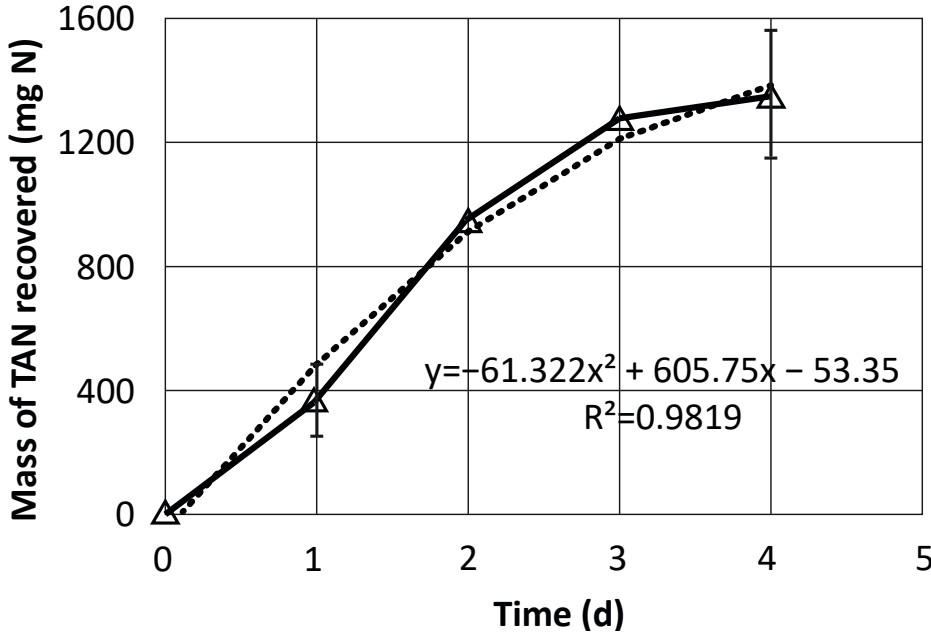

**Figure 4.** Mass of TAN recovered by the acidic trapping solution.

Applying air stripping, a well-known N extraction technology, temperatures above 35 °C pH values higher than 10 are needed to remove more than 50% of N [2,32]. For that, air stripping can present high requirements of energy and chemicals compared with the GPM technology. This technology has been successfully proved to recover NH$_3$ from digestate at pilot scale [3]. Now, the Life Green Ammonia aims at scaling results to the livestock sector market by means of commercial models.

Finally, the treatment also reduced organic matter, solid content, and phosphorous from digestate (Table 1, Section 2.). Thus, for TCOD, TS, and VS, the corresponding removal efficiencies were 35.6%, 28.0%, and 28.7%, respectively. The removal efficiency for Pt was 47.1%. The removal of these compounds could be due to the biological degradation processes that take place at ambient temperature [20].

### 3.2. Agronomic Assessment of the N Recovery Solution

After 34 days of growth, the N fertilisation significantly increased ($p < 0.001$, ANOVA) the yield (fresh matter and dry matter) of the triticale (Figure 5a) compared with the treatments without N fertilisation. However, the triticale fertilised with the ASS was significantly higher compared with the W and C treatments and with the treatment fertilised with the Hoagland solution. The relative yield (RY), evaluated as the proportional increase/decrease in relation to the yield of the Hoag treatment, showed the same trend (Figure 5b). In this case, the RY increase in the ASS treatment ranged between 20% and 29% regarding the RY reported to the fresh matter or the dry matter production, respectively.

**(a)**

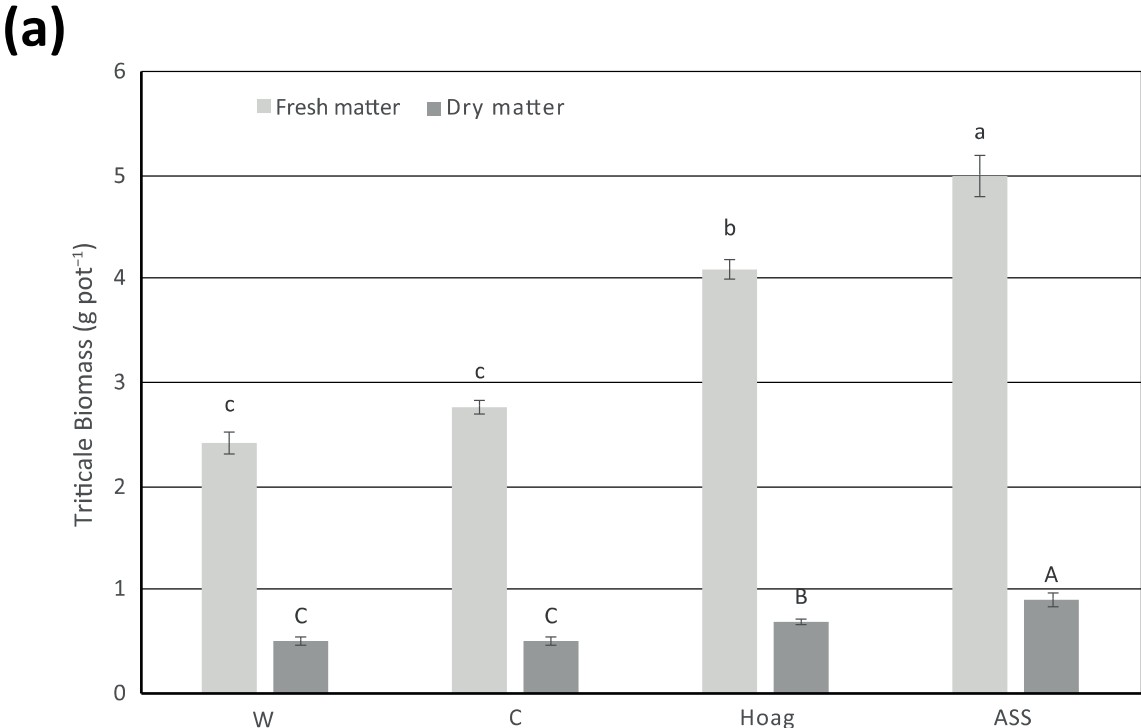

**(b)**

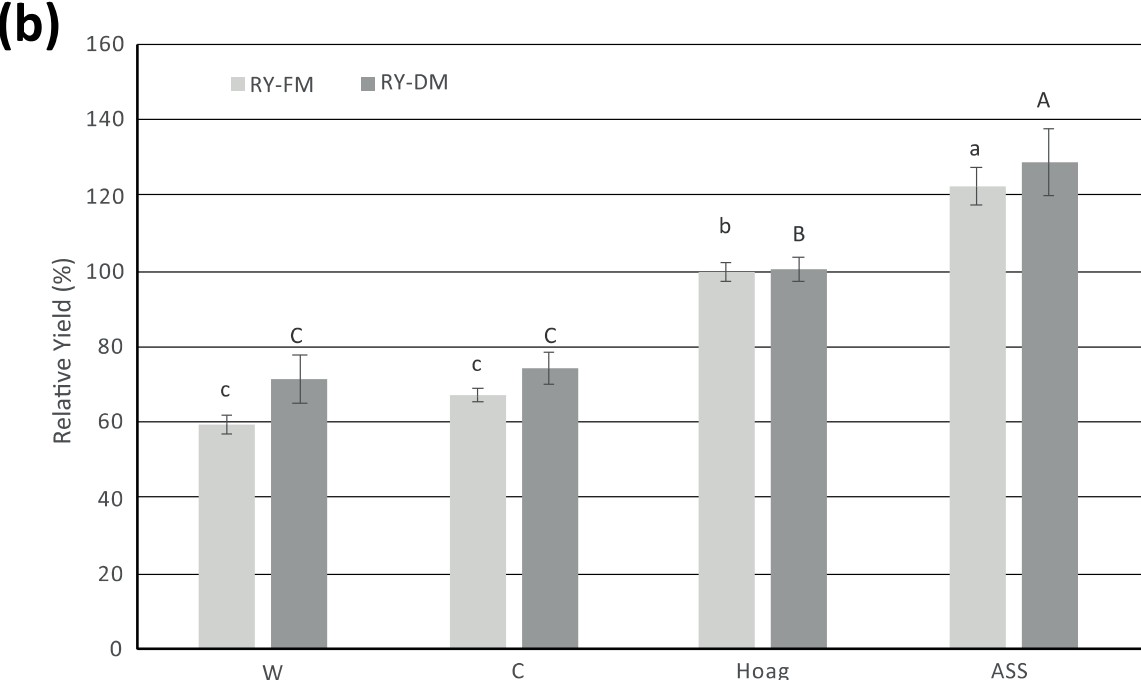

**Figure 5.** Triticale yield after 34 days of growth for each treatment; (**a**) fresh matter and dry matter yield (average ± SE; g pot$^{-1}$) and (**b**) relative yield (average ± SE; %). W—treatment without any fertilisation; C—Control treatment, fertilised with a half-strength Hoagland solution without N; Hoag—treatment fertilised with a half-strength Hoagland solution; ASS—treatment fertilised with a half-strength Hoagland solution without N plus the recovered ammonium sulfate solution. Different letters above the columns show statistical differences between the treatments by Tukey's test. Lower case letters refer to the statistical difference between fresh matter or RY of fresh matter, and capital letters refer to the statistical difference between dry matter or RY of dry matter.

Regarding the N uptake, the treatment fertilised with the ASS showed the highest ($p < 0.001$, ANOVA) value (29.86 mg N pot$^{-1}$), while the Hoagland solution caused a significantly lower N uptake of 24.44 mg N pot$^{-1}$ (Figure 6). The other two treatments with no N fertilisation showed very low N uptake values without significant differences (5.54 and 6.24 mg N pot$^{-1}$ for W and C treatments, respectively).

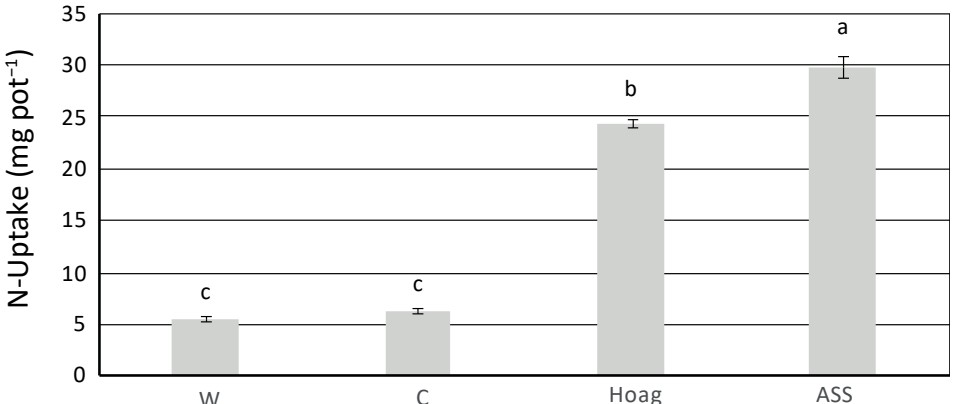

**Figure 6.** N uptake from the triticale during 34 days of growth for each treatment (average $\pm$ SE). W—treatment without any fertilisation; C—Control treatment, fertilised with a half-strength Hoagland solution without N; Hoag—treatment fertilised with a half-strength Hoagland solution; ASS—treatment fertilised with a half-strength Hoagland solution without N plus the recovered ammonium sulfate solution. Different letters above the columns show statistical differences between the treatments by Tukey's test.

As previously mentioned, there are not many works related to the agronomic assessment of the N recovered from bio-wastes. However, Sigurnjak et al. [6,7] reported a pot experiment with lettuce and a field experiment with maize, both fertilised with a recovered ammonium solution obtained from air scrubber water from a pig farm or with a N-fertilised mineral. In this experiment, the lettuce and maize yields and their N uptake in the treatments fertilised with the recovered ammonium solution were similar to those obtained with the mineral N fertilisation. Additionally, Vaneeckhaute et al. [8], with a similar experiment as Sigurnjak et al. [7], reported similar results with a maize crop. An increase in the wheat biomass fertilised with an ASS compared with the mineral ASS fertilisation was reported by Majd [9] in a work with N recovered by GPM technology from liquid dairy manure.

In this work, the N recovery and N agronomic efficiency indices showed that the triticale fertilised with the ASS had significantly higher values (Figure 7a; $p < 0.001$, ANOVA). The triticale used almost 24% of the N applied by the ASS, whereas only 18% was used by the triticale fertilised with the Hoagland solution. The agronomic efficiency of the ASS was significantly higher ($p < 0.001$, ANOVA) and almost double (3.8 mg DM mg$^{-1}$ N) that provided by the Hoagland solution (1.8 mg DM mg$^{-1}$ N; Figure 7b).

Finally, the superior fertiliser value of the ASS regarding the Hoagland solution (Hoag) was highlighted by the ASS treatment indices for the N replacement use efficiency (NRUE), which was 122%, and by the N fertiliser replacement value (NFRV) was 133% (Equations (4) and (5)). In the experiment mentioned above reported by Sigurnjak et al. [7], the NRUE and NFRV were similar in both crops (lettuce and maize) between the treatments fertilised with the recovered ammonium sulphate solution or with the mineral N fertilisation.

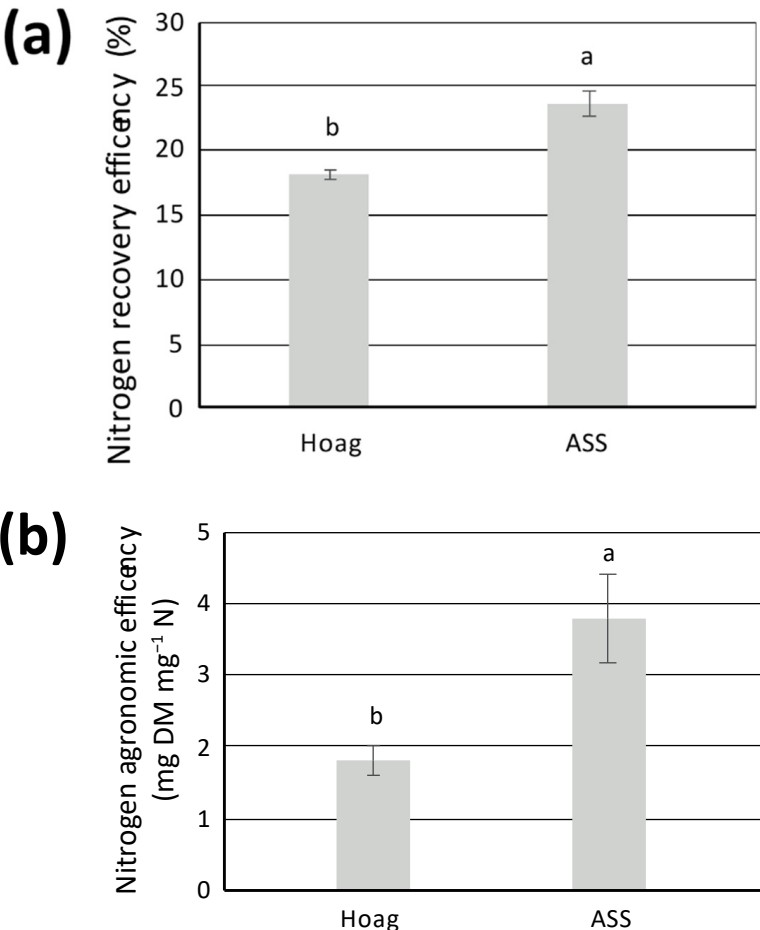

**Figure 7.** N use efficiency indices of the triticale during 34 days of growth for each treatment (average ± SE). (**a**) N recovery efficiency (%) and (**b**) N agronomic efficiency (mg DM mg$^{-1}$ N). W—treatment without any fertilisation; C—Control treatment, fertilised with a half-strength Hoagland solution without N; Hoag—treatment fertilised with a half-strength Hoagland solution; ASS—treatment fertilised with a half-strength Hoagland solution without N plus the recovered ammonium sulfate solution. Different letters above the columns show statistical differences between the treatments by Tukey's test.

The results of this work consistently showed a superior fertilised value of the recovered ASS over the Hoagland solution. The only difference between the treatments (ASS and Hoag) was the N source used. In the case of the Hoagland solution, the N sources were $KNO_3$ and $NO_3NH_4$, and for the ASS treatment, the nitrogen was recovered as ammonium sulphate in solution. Since the fertilisation in both treatments was carried out with all the nutrients needed for crop growth, the differences observed between them can only be explained by the composition of the recovered ASS. The reported results from the trials of other authors [6–8] were obtained with air scrub water from livestock farms. In this work, the liquid fraction from the anaerobic co-digestion of orange peel and swine manure was used to obtain the ammonium sulphate solution. As referred to previously, the composition of the orange peel bio-waste is well studied [22–25] and contains tannins, phenolic compounds, carotenoids, dietary fibres, sugars, and organic acids (e.g., oxalic, citric, acetic and succinic), making this bio-waste valuable for the recovery of bioactive compounds with positive health aspects. The anti-oxidant activity of the orange peel bio-waste is about 90.25% [23]. However, during the anaerobic digestion of the orange peel biowaste used in this work, the temperature was in the mesophilic range of 37–38 °C, which can have some adverse effects on these bioactive compounds.

Nevertheless, Garau et al. [22] reported that temperatures between 50 and 60 °C promoted minor disruption of cell wall polymers, particularly pectic substances, the anti-oxidant capacity remaining stable in the range of 40–70 °C. The ascorbic acid seems to be very unstable, showing easy chemical oxidation, but the carotenoids are very heat stable compounds [24]. The results of Riaño et al. [20] and García-González et al. [21] about the gas-permeable membrane technology to recover N applied to a liquid centrifuged swine manure showed that some organic matter of the manure was recovered in the acidic solution. This work also suggests that during the process of the N-recovery, some bioactive compounds may have passed through the membrane and have a positive effect on plant growth. Indeed, the TCOD of the ASS was 88 mg $L^{-1}$. The same trend on plant growth was observed by Tuttobene et al. [33], reporting that the use of dried orange wastes in the fertilisation of durum wheat (*Triticum durum* Desf.) promotes plant vigour (Leaf Area Index, LAI, and above-ground biomass) much more than the mineral fertilisation with the observation that the crop growth was higher, by up to +400% with the organic fertilisation compared with the mineral fertilisation.

Raman spectroscopy measures the inelastic scattering that results in a characteristic vibrational spectrum related with various functional groups present in a sample. Thus, it can be useful to discriminate samples composition. As can be observed in Figure 8, the spectra obtained from the present studied liquid fertilisers (ASS, Hoag, C) ranged between 1800 $cm^{-1}$ and 700 $cm^{-1}$. However, the spectra were collected between 4000 $cm^{-1}$ and 200 $cm^{-1}$, but only the above-mentioned regions presented relevant discriminant information.

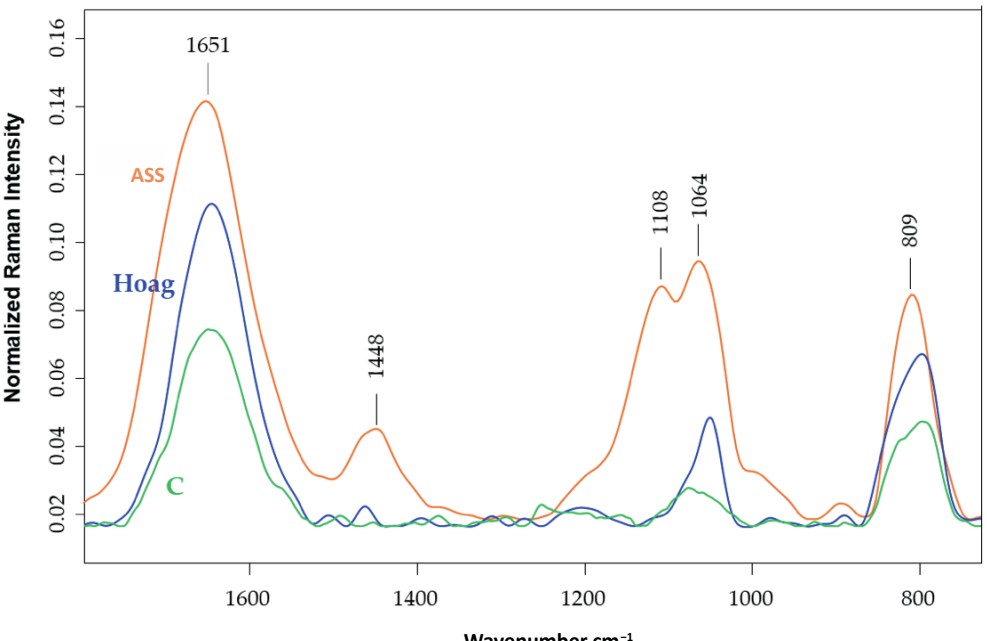

**Figure 8.** Raman spectra of aqueous fertiliser solution: C—Control treatment, fertilised with a half-strength Hoagland solution without N; Hoag—treatment fertilised with a half-strength Hoagland solution; ASS—treatment fertilised with a half-strength Hoagland solution without N plus the recovered ammonium sulfate solution. The spectrum was obtained over three measurements and the baseline correction was made.

As previously mentioned, the liquid fraction from the anaerobic digestion of orange peel should be rich in tannins, phenolic compounds, carotenoids, dietary fibres, sugars, and organic acids, and as it was expected that the Raman signal would comprise the functional groups of this bioactive compounds. This suggests that some of them should be passed to the ASS.

The peak at 1651 cm$^{-1}$ was assigned to N–H primary amine, and NH and OH bonds. These links could be found in the C and Hoag fertilisers solutions but in more concentration in the ASS solution and could be easy to distinguish in the FT-Raman spectra (Figure 8). The spectral signal around 1448 cm$^{-1}$ could be also assigned to the N–H bonds but also to some combination of the vibration of COO- group and CH$_2$ bending vibration [34–36]. This region was usually attributed to the presence of organic acids and flavanols in the analysed matrix [35]. Because of that, this region appears only in the ASS fertiliser solution.

Regarding the most discriminated region between 1200 cm$^{-1}$ and 1000 cm$^{-1}$, a higher differentiation between the N-rec fertiliser solution and the other ones was observed. Huang et al. [37] identified the Raman signal around 1046 cm$^{-1}$ attributed to nitrate symmetrical stretching by using calcium nitrate (Ca(NO$_3$)$_2$), potassium nitrate (KNO$_3$), and ammonium nitrate (NH$_4$NO$_3$) standard chemicals. This region is assigned to the symmetric stretching of the three oxygen atoms of the nitrate ion [37]. In this case, it is easy to observe that N-rec solution is richer than the other ones (Figure 8). In addition, according to Kizil et al. [34], this region was assigned to the carbohydrates bending vibration of C–H and C–O–H and by the proteins and amino acids vibration of the C–N bond in amino acids and proteins.

The C–O–C linking could be observed also in this spectral region and could be associated to some organic compounds in the ASS which was recovered from the anaerobic digestate. The peak at 1108 cm$^{-1}$ could be assigned to the C–OH deformation, a functional group related to the compound present in the orange peel, as well to a combination of stretching vibration of C–O and C–O–C and also to the vibration of C–N of protein and amino acids already observed at 1064 cm$^{-1}$ [21].

At 809 cm$^{-1}$ it was possible to observe the signal of the C–H and CH$_2$ vibration [34], C–O–H bending [38], and of the C–Cl stretch of the aliphatic chloro compounds. So, the Raman spectra confirms the presence of chemical bonds in the ASS compatible with the presence of organic compounds, namely proteins, carbohydrates, and some bioactive compounds. These organic compounds can provide a biostimulant effect on triticale and justify the higher yield and the higher N use efficiency indices of the triticale fertilised with ASS compared with Hoag and C treatments. Biostimulants are defined as organically based plant-promoting substances/microorganisms applied to soil to increase the nutrient uptake, stimulate plant growth, increase tolerance to abiotic and biotic challenges, and improve product quality [39]. Several works with vegetable crops and medicinal crops showed that the use of biostimulants increases plant growth by increasing nutrient availability and thus nutrient plant content and antioxidant enzyme activity in the plants [39–41], and in the medicinal plants also increased the synthesis of the secondary compounds of medicinal interest [42].

This ammonium sulphate solution was shown to be a valuable liquid fertiliser to be recognised as "EC fertilizer—category CI b) (i) in the current Fertilizer Regulation EU 2019/1009 [43]. However, future work is needed to clarify the complete composition of the recovered ASS and confirm its superior agronomic effect in the field.

## 4. Conclusions

With this work, it is possible to conclude that the application of the gas-permeable membrane technology for the recovery of N from anaerobically digestate with the addition of orange peel proved to be very efficient with an increase of TAN recovery rate of 30% to 50% higher than that observed in other works using only anaerobic digestate from swine manure.

This work also showed that the ASS solution obtained had a higher fertilising value compared with the mineral fertilisation. This result suggests that some organic compounds recovered in the ASS can have a bioestimulant effect on plant growth.

The Raman spectroscopy technique allowed for the identification of the chemical composition of ASS, as well as the identification the more promising regions to charac-

terise/discriminate the organic compounds in the ASS obtained by the gas-permeable membrane technology early.

However, further studies are needed to better understand the chemical mechanisms of these complex matrices and their effective effect on plant growth in the field.

**Author Contributions:** Conceptualization, C.H., B.R. and M.C.G.-G.; formal analysis, C.H.; investigation, C.H. and B.R.; data curation, C.H. and B.R.; O.A. Raman analysis and interpretation. writing—original draft preparation, C.H. and B.R.; writing—review and editing, C.H., B.R. and M.C.G.-G.; project administration, C.H. and M.C.G.-G.; funding acquisition, C.H. and M.C.G.-G. All authors have read and agreed to the published version of the manuscript.

**Funding:** This work has been funded by the National Institute of Research and Agro-Food Technology (INIA) and co-financed with FEDER funds (PID2019-106148RR-C41) and for the EU Program INTERREG V-A Spain—Portugal (POCTEP) 2014–2020 (Project 0745_SYMBIOSIS_II_3_E). This work has been also funded by National Funds through FCT—Foundation for Science and Technology under the Projects [UIDB/00681/2020] [CERNAS-IPCB] and UIDB/00239/2020 [CEF].

**Institutional Review Board Statement:** Not applicable.

**Informed Consent Statement:** Not applicable.

**Data Availability Statement:** The data that support the findings of this study are available upon reasonable request.

**Conflicts of Interest:** The authors declare that they have no conflict of interest.

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
