# Peer review of "Fertiliser Effect of Ammonia Recovered from Anaerobically Digested Orange Peel Using Gas-Permeable Membranes"

_sustainability, doi:10.3390/su14137832_

Round 1

Reviewer 1 Report

I reviewed the manuscript entitled " Fertilizer effect of ammonia recovered from  anerobically digested orange peel using gas permeable membranes".  The manuscript has a lot of information. In my opinion, the paper is very well written and acceptable for publication. The manuscript needs some correction before it can be published. I suggest authors take a closer look and adjust the write up to be more precise and appealing to the readers.

1.       It is better to do not to use the first person's pronoun. Do not use "we, us, or our" throughout the paper.

2.       The English are generally OK.

3.       The percentage purity and company of all reagents/chemicals utilized must be reported.

Author Response

Answers to the Reviewer 1 –

The authors thank the valuable suggestions done by the reviewer in order to improve the manuscript.

The modifications done in the manuscript were highlighted in yellow colour.

Comment 1: “It is better to do not use the first person´s pronoun. Do not use “we, us or our” throughout the paper”.

 The manuscript was revised and the phrases with the pronouns “we us, our” were rephrased.

Comment 2: “The English are generally OK”.

Comment 3: “The percentage purity and company of all reagents/chemicals utilized must be reported”.

The reagents had a percentage of purity ³ 98%.  The H2SO4 was Panreac (line 92) and the others reagents were Merck. The information was included in lines 182-183

The authors remain available for further improvements in the manuscript.

Reviewer 2 Report

NONE

Author Response

Answers to the Reviewer 2 -

The manuscript was improved in the introduction, a hypothesis was included, references were also included and the conclusions were modified.

The modifications done in the manuscript were highlighted in yellow colour.

The authors remain available for further improvements in the manuscript.

Reviewer 3 Report

The research work is very interesting and valuable, and the authors have also obtained good research results. But there are still some issues in the manuscript, when they feel appropriate, I very kindly advise the authors to consider and address them to improve the quality of the manuscript. 

Please see my comments and suggestions below:

L87: Please add notes to figure 1.

L96: How many days is the digestate period?

L104: Please describe how many times the experiment was duplicated?

L129: How much nitrogen content is in the soil?

L136: Why is the N fertilization was divided into 18 applications? What is the amount of each irrigation?How much N fertilization is applicated each time? Please rephrase it.

L198: Why did the authors choose these 4 days for the purpose of this study? Please briefly justify it in the text.

L208, 211: Please delete the background line in Figure 2 and Figure 3.

L211: Why is there no standard error in the value of TAN concentration in Figure 3.

L215:A format is wrong here. Please check it.

L215: Why is the TAN recovery rateof this study so high (43.0 ± 6.6 g N per m-2d-1), about twice that of previous studies (22.7 and 30.7 g N m-2 d-1)? Please discuss the possible reasons

L229-230: Please give the citations for the sentence.

L233: This sentence: “Tukey's test was used to compare means at a 0.05 probability level” in 2.4. The authors included the term “P<0.05” in the 2.4, but whythe resultsof yield are significantly at P <0.001. Please clarify these aspects.

L242: I suggest the authors delete the table (Fresh matter and Dry matter), the same below.

L244:The error bars are inconsistent (degree of thickness) in Figure 5. Please use the drawing software to generate them automatically, the same below.

L284: Please give the citations for the sentence. Please further explain why it is negative value. Is the nitrogen content very low in the soil?

L309: A word is wrong here. Please check it.

L398-401: This paragraph reads more like Discussion than Conclusion.

L400: A format is wrong here. Please check it.

Author Response

Answers to the Reviewer 3

 The authors thank the valuable suggestions done by the reviewer in order to improve the manuscript.

The modifications done in the manuscript were highlighted in yellow colour.

Comment 1: “L 87: Please add notes to Figure 1.”

The explanation of the Figure 1 has been included in the manuscript (Lines 107-114):

“Figure 1. Experimental set-up of the gas-permeable membrane system, consisting of 1) a wastewater vessel, 2) a tubular membrane submerged in the wastewater, 3) a tank containing the acidic trapping solution, 4) a peristaltic pump that continuously recirculated the acidic solution through the tubular membrane, 5) an air pump to supply air to wastewater and 6) an airflow meter to control the supplied airflow rate. “

Comment 2. “L 96: How many days is the digestate period?”

The anaerobic digestion at batch conditions lasted 73 days. This information has been included in the manuscript (L117-119):

“…it was produced during the anaerobic co-digestion of orange peel with swine manure in mesophilic digesters operating at batch conditions for 73 days.”

Comment 3: “L104: Please describe how many times the experiment was duplicated”.

As suggested, it has been specified in the manuscript the times the experiment was duplicated (Line 130):

“The experiment was duplicated twice…”.

Comment 4: “L129: How much nitrogen content is in the soil?”.

The content of the total nitrogen (Nt) in the soil was 2.96 g kg-1. The information was added to the manuscript (Line 157).

Comment 5: “L136: Why is the N fertilization was divided into 18 applications? What is the amount of each irrigation? How much fertilization is applicated each time? Please rephrase it.

In order to provide a continuous nitrogen supply to crop, the fertilization was carried out continuously, spread over 18 applications. If we apply all the nitrogen in the beginning of the experiment, the risk of the increase the soil salinity is very high. So, spread the N fertilization through the period of the experiment avoid also the negative effects of soil salinity on crop development.

In each fertilization we applied 13 ml/pot of the Hoagland solution or 3.3 ml/pot of the ASS solution.  Then we watered with deionized water all the pots until 75% of the field capacity. We control the amount of deionized water applied in each pot by the weight of the pots.

In each time the N fertilization applied was 0.1g/pot divided by 18 applications. So, the amount of N applied in each N fertilization/pot was 5.56 mg N/pot.

The paragraph was rephrased (Lines 162-168): “In order to provide a continuous nitrogen supply to crop, the fertilization was carried out continuously through the period of the experiment. So, the N fertilization was spread over 18 applications to obtain at the end of the experiment a total amount of 100 mg N per pot. In each N fertilization it was applied 13 ml pot-1 of the Hoagland solution, and 3.3 ml pot-1 of the ASS solution. Then all pots were watered with deionized water until 75% of the field capacity. The control of the amount of deionized water applied was done by the weight of the pots”. 

Comment 6: “L198: Why did the authors choose these four days for the purpose of this study? Please briefly justify in the text. 

The end of the experiment was determined by the marked decreased observed in the mass of TAN recovered by the gas-permeable membrane system in the day 4 of the experiment, as can be seen in Fig. 4. A brief explanation has been included in the manuscript (Line 245):

“Specifically, this occurred after 3rd day of experiment”

Comment 7: “L208, 2011: Please delete the background line in Fig. 2 and Fig. 3”.

The background lines in Fig. 2 and Fig.3 have been deleted as suggested.

Comment 8: “L211: Why is there no standard error in the value of TAN concentration in Fig. 3”.

The standard errors are represented in Fig. 3 for TAN concentration in digestate. However, they are so small (from 1 to 24 mg L-1) that they cannot be appreciated in the Figure.

Figures 2 and 3 were rearranged Lines 237 and 239.

Comment 9: “L215: A format is wrong here. Please check it.”

The format has been checked and corrected as suggested.

Comment 10: “L215: Why is the TAN recovery rate of this study so high (43.0 ± 6.6 g N per m-2 d-1) about twice that of previous studies (22.7 and 30.7 g m-2 d-1). Please discuss the possible reasons.”

The possible reasons that can be determined the higher TAN recovery rate obtained in the present study has been discussed in the manuscript (L252-256):

“The higher TAN recovery rate obtained in the present study could be attributed to the high pH (near 9.50) achieved in the digestate, that it is the most critical parameter determining the amount of free NH3 to pass through the membrane [31 - García-González and Vanotti, 2015]. Other factors that can influence in the TAN recovery rate by these membrane systems are the initial TAN concentration and the mixing [[31] - García-González and Vanotti, 2015; [2] - Munasinghe -Arachchige et al., 2020 [2]].”

Comment 11: “L229-230: Please give the citations for the sentence”.

The reference “[20]” ( Riaño et al., 2019) has been included, as proposed (L271).

Comment 12: “L233: This sentence “Tukey´s test was used to compare means at a 0.05 probability level” in 2.4. The authors included the term “P <0.05” in the 2.4, but why the result of yield are significantly at P <0.001. Please clarify these aspects.”

The level of significance reported in line 233 274 (P <0.001) is referred to the ANOVA analysis. The ANOVA gives the statistical information about the effect of the fertilization on the biomass production at P <0.001 or P < 0.01 or P <0.05 or not significant.

The Tukey test compare the significant differences between the treatments. This comparison is done at a probability of P <0.05.

The reference to ANOVA was included near P <0.001 (Lines 274; 296; 321; 324).

Comment 13: “L242: I suggest the authors delete the table (Fresh matter and dry matter), the same below.”

The table below the Figures was removed: Fig. 5, Fig. 6 and Fig. 7. (Lines 282; 300; 326)

Comment 14: “L244: the error bars are inconsistent (degree of thickness) in Figure 5. Please use the drawing software to generate then automatically, the same below.

The bars were redrawing in Fig. 5, Fig. 6 and Fig. 7 (Lines 282; 300; 326)

Comment 15: “L284: Please give the citations for the sentence. Please, further explain why it is negative value. Is the nitrogen content very low in the soil?”

The N recovery efficiency and the N agronomic efficiency (Equations 1 and 2, Lines 189 and 191) were calculated with the values of the Control treatment for the N0uptake and for the DMY0.

In the first version of this manuscript these equations were also applied for the W treatment to highlight the low amount of N provided by the soil and the relevance of the N fertilization for the crop. However, we understand that the inclusion of this treatment (W) can be confusing and not necessary to display in the manuscript, since the W treatment is very similar to the Control. Both (C and W) had no nitrogen fertilization. So, the W treatment was removed from the Figure 7 and from the text (Lines 322-325).

Comment 16: “L309: A word is wrong here. Please check it.”

The phrase was rewritten, Lines 349-350: “the N sources were KNO3 and NO3NH4, and for the ASS treatment, the nitrogen recovered as ammonium sulphate in solution”

Comment 17: “L398-401: This paragraph reads more like discussion than conclusion”.

The conclusions have been modified according to Comment 6 of Reviewer 4:

“4. Conclusions

With this work it is possible to conclude that the application of the gas-permeable membrane technology for the recovery of N from anaerobically digestate with the addition of orange peel proved to be very efficient with an increase of TAN recovery rate of 30% to 50% higher than that observed in other works using only anaerobically digestate from swine manure.

This work also showed that the ASS solution obtained had a higher fertilizing value compared with the mineral fertilization.  This result suggests that some organic compounds recovered in the ASS can have a bioestimulant effect on plant growth.

The Raman spectroscopy technique allowed for the identification of the chemical composition of ASS, as well as identify the more promising regions to characterize/discriminate the organic compounds in the ASS obtained by the gas-permeable membrane technology early.

However, further studies are needed to better understand the chemical mechanisms of these complex matrices and their effective effect on plant growth in the field.”

Comment 18: “L400: A word is wrong here. Please check it.”

The conclusions have been modified according to Comment 6 of Reviewer 4.

The authors remain available for further improvements in the manuscript.

Reviewer 4 Report

Overall, I find the article well written and helpful. Despite this, I have a few remarks.

1)     Be more specific in the Abstract (give particular results, for example, numerically).

2)     There needs to be a better introduction into the problematics.

3)     I suggest you to add hypotheses in introduction.

4)     What are the novelties of this paper?

5)     The ordinate title of Figure 7 is not clear, please modify it.

6Conclusion is not a description of simple results, it needs to be summarized and refined.

Author Response

Answers to the Reviewer 4 –

The authors thank the valuable suggestions done by the reviewer in order to improve the manuscript.

The modifications done in the manuscript were highlighted in yellow colour.

Comment 1: “Be more specific in the Abstract (give particular results, for example, numerically)”.

The abstract was modified and some numerically results were added to the abstract: L25

Comment 2: “There needs to be a better introduction into the problematics”.

The introduction has been improved, deepening in the need of searching new sources of nitrogen to be used as fertilizer as well as the problematic of the use of digestate in nitrate-vulnerable zones (L40-45):

“In this sense, the recovery of N from wastewater could partially offset the demand for nitrogen-based fertilizers [2]. The effluents produced after anaerobic digestion of agroindustrial waste (digestate) are an important source of N. At the same time, decreasing the N content form anaerobic digestate can reduce the environmental risks associated with its use as fertilizer in nitrate-vulnerable zones [3].”

Comment 3: “I suggest you to add hypothesis in introduction”.

A hypothesis was included: Lines 65-82.

“….from the GPM technology applied to an anaerobic digestate provided by the co-digestion of an orange peel effluent mixed with swine manure. To the best of our knowledge, this is the first work in which the agronomic behaviour of the ASS from anaerobic digestate was tested that included also, the orange peel in its composition. Indeed, the agronomic role of digestates from the anaerobic digestion are well known, not only acting as soil amendments increasing the level of the soil organic matter (SOM) [10–14] but also their role as a source of nutrients to crops [15–19]. However, the GPM techonology can constitute another approach in order to obtain a bio-based liquid fertilizer with a known N concentration and with commercial value. Regarding the composition of the ASS Riaño et al. [20] and García-González et al. [21] observed that some organic matter can also be recovered by the GPM thechnology. So, the presence of organic compounds in the ASS otained in this work can also be expected. However, the composition of the orange peel bio-waste can add value to the digestate and to the ASS since it contains tannins, phenolic compounds, carotenoids, dietary fibres, sugars and organic acids (e.g. oxalic, citric, acetic and succinic) making this bio-waste valuable for the recovery of bioactive compounds [22 –25]. So, this work hypothesised that the GPM technology in addition to recover N from the anaerobic digestate can also recovered some organic compounds which can contribute to the improvement of the fertilizing value of the ASS.”

Comment 4. “What are the novelties of this paper?”

The novelties of this work were: This work tested for the first time an anaerobically digestate which include an orange peel bio-waste for (i) the N recovery by the gas-permeable membrane and (ii) the agronomic behaviour of the ammonium sulphate solution obtained. Also, the RAMAN spectroscopy was assessed to identify the chemical composition of the ASS solution to the early characterization of its organic compounds.

Comment 5: “The ordinate tittle of Figure 7 is not clear, please modify it”.

Fig. 7 was rearranged, L326.

Comment 6: “Conclusions is not a description of simple results, it needs to be summarized and refined”.   

Conclusions were modified:

“4. Conclusions

With this work it is possible to conclude that the application of the gas-permeable membrane technology for the recovery of N from anaerobically digestate with the addition of orange peel proved to be very efficient with an increase of TAN recovery rate of 30% to 50% higher than that observed in other works using only anaerobically digestate from swine manure.

This work also showed that the ASS solution obtained had a higher fertilizing value compared with the mineral fertilization.  This result suggests that some organic compounds recovered in the ASS can have a bioestimulant effect on plant growth.

The Raman spectroscopy technique allowed for the identification of the chemical composition of ASS, as well as identify the more promising regions to characterize/discriminate the organic compounds in the ASS obtained by the gas-permeable membrane technology early.

However, further studies are needed to better understand the chemical mechanisms of these complex matrices and their effective effect on plant growth in the field.”

The authors remain available for further improvements in the manuscript.

Reviewer 5 Report

I reviewed the manuscript titled Fertilizer effect of ammonia recovered from anaerobically 2 digested orange peel using gas-permeable membranes by Carmo Hortaet al. In general, the manuscript meets the characteristics of a research paper. Some minor and major details need to be taken care of before being published. - I suggested that the authors compare the results of membrane N extraction to another technology used for N recovery from digestate? - This technology and processes proposed in this study, can be continuously developed at industrial scale in coupling to anaerobic digestion? If yes, you can give some example and/or recommendations - Figure 7: some problem on the X axis (N agronomic efficiency?), I suggested to replace N by Azote and correct the unit, C-control, not presented in the figure 7? - Some problem with nomenclature, I suggested that the author use a simple nomenclature in the text and legend of all figures - What the effect of the digestate on the plant growth without N extraction? You can compare these results to an industrial fertilizers rich in N in the Figures 5-7 - I suggest to cited some reference about the use of digestat as a biofertilizers · Tayibi et al. 2021. Science of The Total Environment 793, 14846 · Tayibi et al. 2021. Journal of Environmental Management 279, 111632

Author Response

Answers to the Reviewer 5 –

The authors thank the valuable suggestions done by the reviewer in order to improve the manuscript.

The modifications done in the manuscript were highlighted in yellow colour.

Comment 1: “I suggest to the authors to compare the results of membrane N extraction to another technology used for N recovery from digestate”.

A comparison with air stripping, a well-known N extraction technology, has been included in the manuscript (L258-263):

“Applying air stripping, that it is a well-known N extraction technology, temperatures above 35ºC o pH values higher than 10 are needed for removing more than 50% of N [2, 32]. For that, air stripping can present high requirements of energy and chemicals compared with the GPM technology.”

Comment 2: “This technology and processes proposed in this study, can be continuously developed at industrial scale in coupling to anaerobic digestion? If yes, you can give some example and/or recommendations.

One of the research line of the Unit of Environmental Technology of the Agroindustrial Technological Institute of Castilla y León is to scale up the gas-permeable technology to recover nitrogen from lab to industrial scale. For that, we participated in the LIFE Ammonia Trapping project, with the aim of testing this technology at pilot scale in an anaerobic digestion plant and in a swine manure farm. This project demonstrated at on-farm pilot plant scale that the GPM technology successfully captures NH3 from manure (Molinuevo-Salces et al., 2020) and from anaerobic digestate (Riaño et al., 2021). Based on this previous experience, we are now participating in the new LIFE Green Ammonia project, that started on October 2021 and aims at escalating results to the livestock sector market by means of commercial models and exploitation plans. This information has been included in the manuscript (L261-263):

“This technology has been successfully proved to recover NH3 from digestate at pilot scale [3]. Now, the Life Green Ammonia aims at scaling results to the livestock sector market by means of commercial models.”

Comment 3: “Figure 7: some problem on the X-asis (N agronomic efficiency?), I suggested to replace N by Azote and correct the unit, C-control, not presented in the figure 7?

Figure 7 was modified, Line 326

Comment 4: “Some problem with nomenclature, I suggested that the author use a simple nomenclature in the text and legend of all figures”.

The nomenclature used was simplified using for the N-rec treatment always the acronym ASS.

Comment 5: “What the effect of the digestate on the plant growth without N extraction? You can compare these results to an industrial fertilizer rich in N in the Figures 5-7.

Comment 6: “I suggest to cited some references about the use of digestate as a biofertilizer. Tayidi et al. 2021 Science of the Total Environment, 793, 14846. Tayibi et al. 2021. Journal of Environmental Management 279, 111632.”

In this work we compared the N fertilization between (i) the solution with the nitrogen recovered from an anaerobic digestate and a (ii) solution done with mineral nitrogen which can be similar to the N mineral fertilization (N industrial fertilizers rich in N) and also with (ii) no nitrogen fertilization. So, for the objectives of this work we considered that the discussion should be done with the works that used similar approaches (e.g. Sigurnjak et al.). Nevertheless, we introduce a paragraph in the Introduction with the references provided by this reviewer (Tayidi et al.), and also with another references, to discuss the use of digestate as a biofertilizer vs ASS (References 10-109, Introduction Lines 68-73).

The authors remain available for further improvements in the manuscript.
